# Retrocardiac Pneumomediastinum: Description of an Unusual Case and Review of Literature

**DOI:** 10.3390/children10040649

**Published:** 2023-03-30

**Authors:** Federica Porcaro, Alessandro Onofri, Annalisa Allegorico, Paolo Tomà, Renato Cutrera

**Affiliations:** 1Pediatric Pulmonology and Cystic Fibrosis Unit, Bambino Gesù Children’s Hospital, IRCCS, 00165 Rome, Italy; 2Imaging Unit, Bambino Gesù Children’s Hospital, IRCCS, 00165 Rome, Italy

**Keywords:** air leak, retrocardiac pneumomediastinum, child, non-invasive ventilatory support

## Abstract

Retrocardiac pneumomediastinum (RP) is the consequence of air trapping in the inferior and posterior mediastinum. It is characterized by the presence of a right or left para-sagittal infrahilar oval or pyramidal air collection on a chest X-ray. It is usually detected in neonates because of alveolar rupture after invasive ventilation or invasive manoeuvres applied on airways or the digestive tract. A healthy child came to the emergency department (ED) for acute respiratory failure due to viral bronchiolitis when he was 2 months old. Because of his clinical condition, he underwent helmet continuous positive airway pressure (HCPAP). When the condition allowed, he was discharged and sent home. He was re-admitted into the hospital for asthmatic bronchitis 3 months later. A frontal chest X-ray taken during the second hospitalization showed an oval-shaped retrocardiac air lucency not previously detected. Differential diagnosis including digestive and lung malformations was made. Finally, the diagnosis of RP was made. We report an unusual case of retrocardiac pneumomediastinum in a 5-month-old male infant after the application of continuous positive pressure via a helmet. RP presentation after the administration of non-invasive ventilatory support beyond the neonatal age is unusual. Although surgical drainage is curative, conservative treatment can be considered in hemodynamically stable patients.

## 1. Introduction

Pneumomediastinum is a clinical entity belonging to air leak syndromes. It can be classified as spontaneous or secondary. Spontaneous pneumomediastinum is generally reported in young age, as well as after respiratory viral infections [1,2,3]. The explanation of why pneumomediastinum presents in young patients is that the mediastinal tissues in young people are looser and softer than their older counterparts, making air progression more difficult to occur [4]. Indeed, when it occurs, it is more frequent in male adolescents during an asthmatic exacerbation [5], after exercise, vomiting, or other Valsalva manoeuvres [6].

Secondary pneumomediastinum can be classified as iatrogenic, traumatic, or non-traumatic.

Chest or abdominal surgery, endoscopic procedures on airways or the oesophagus, pleural cavity instrumentation, vascular access procedures, and mechanical ventilation are common causes of iatrogenic pneumomediastinum [7]. The latter acts through the mechanisms of barotrauma, as described below [4,8].

While traumatic pneumomediastinum is due to blunt injuries and penetrating chest or abdominal injuries, non-traumatic pneumomediastinum often occurs in patients with underlying diseases such as air trapping, asthma, chronic obstructive pulmonary disease (COPD), bronchiectasis, interstitial lung disease, inhalation of toxic fumes, and malignancy [1].

The term pneumomediastinum defines the presence of air in the mediastinal space after the rupture of bronchioles, alveolar ducts, or alveoli.

Iatrogenic pneumomediastinum in the paediatric population usually affects term or preterm neonates who are particularly vulnerable to air leaks because of respiratory distress, hyaline membrane disease, and meconium aspiration [9]. It is the third most common type of air leak in neonates, preceded by pulmonary interstitial emphysema and pneumothorax [10].

It is well known that the mediastinum is the internal space of the ribcage, bounded laterally by the pleural cavities and the lungs, anteriorly from the sternum, and posteriorly from the vertebral column. It can be divided into the superior and inferior mediastinum. The latter is further divided into the anterior, middle, and posterior mediastinum [10]. Posterior or retrocardiac pneumomediastinum (RP) is the effect of the passage of air along the sheath of the perivascular connective tissue towards the hilum and its accumulation in the space between the pericardium and the vertebral spine. This condition has long been described in term (≥37 weeks of gestational age) or preterm (<37 weeks of gestational age) newborns who have undergone intensive care and invasive ventilatory support (Table 1).

Symptom presentation depends upon the entity of air accumulation. Coughing, vomiting, and chest or neck pain are symptoms often reported by older patients who are able to report them [4]. However, signs of respiratory distress such as restlessness, tachypnoea, thoracic retractions, and increase in oxygen requirements can be present depending on the size of pneumomediastinum [3]. More severe clinical pictures caused by a greater accumulation of air in the mediastinal space are due to vessels and tracheal obstruction that are responsible for cardiac tamponade and decreased venous return.

Generally, an anteroposterior (AP) chest X-ray is sufficient to make a diagnosis of pneumomediastinum, although a lateral view may add useful information in unclear cases. In particular, the radiological presentation of RP is defined by the presence of right or left para-sagittal infrahilar oval or pyramidal lucent collection [10].

A chest computed tomography (CT) scan is a useful diagnostic instrument to assess the extent of air collection, to confirm the diagnosis in cases with an inconclusive chest X-ray, and to identify causative factors or other air leak conditions (pneumothorax, pneumopericardium, or pneumoperitoneum) [4].

Bronchoscopy, oesophagoscopy, or oesophagography are not routinely required, except when airway or digestive tract injuries are suspected [4,22].

Pneumomediastinum is commonly considered a benign entity. Indeed, mild forms of pneumomediastinum resolve on their own and require no invasive interventions. In these cases, treatment is directed towards symptom relief (analgesics or oxygen administration to improve pain and decrease respiratory fatigue, respectively), and serial chest X-ray may be used to monitor the evolution until pneumomediastinum resolution [10]. However, some patients may develop malignant pneumomediastinum, secondary pneumothorax, or pneumoperitoneum undermining clinical stability, and therefore require prompt surgical drainage [18,19,20,21].

We report an unusual case of retrocardiac air collection in a 5-month-old male infant without any history of neonatal resuscitation, positive ventilation, or other invasive manoeuvres on the airway or digestive tract in the first month of life. Parental consent to description and publication of the paper has been collected.

## 2. Case Description

A 5-month-old male infant was first evaluated at our Respiratory Unit for dry cough for a few days. He was born at 38 gestational weeks from a spontaneous delivery. The ante-natal and post-natal periods were normal, and respiratory distress at birth was not reported. He was breastfed and growth was regular. He was hospitalized at 2 months of life with bronchiolitis due to respiratory syncytial virus (RSV) infection. Upon admission, the physical examination revealed the presence of bilateral wheezing, and a chest X-ray showed a parenchymal hypodiaphania in the right upper lobe, a bilateral widespread increase in air content, and peribronchial thickening (Figure 1a) that were all attributed to the current infection. In the following hours, the baby developed acute respiratory failure with fast breathing (respiratory rate 95/min), nasal flaring, rib retractions, and feeding refusal. The blood oxygen level on pulse oximetry fell to around 92% or lower, and carbon dioxide levels in capillary blood were 65 mmHg. Due to the progressive worsening of clinical conditions, minimal sedation with dexmedetomidine was started, and helmet-CPAP 40 L/min, PEEP pressure 7 cmH_2_O, and O_2_ supplementation (FiO_2_ 0.5%) was administered for 48 h, with progressive reduction in pressure and oxygen levels in the following three days based on the improvement of clinical condition. He was discharged 10 days later when the general condition and caloric intake became satisfactory.

He returned to our facility with dry cough for a few days when he was 5 months old. The baby was in generally good condition with RR 50/min, heart rate 160/min, spO_2_ levels 96%, and normal carbon dioxide levels in capillary blood. Chest inspection showed mild rib retractions and widespread wheezing. A new viral infection sustained by Rhinovirus was detected with viral investigation carried out on nasal swab, and the diagnosis of asthmatic bronchitis was made. In view of the poor response to inhaler bronchodilator and systemic steroid treatment, the baby underwent a new chest X-ray (AP and lateral view) that showed circular aerial content with a right paramedian location in the posterior mediastinum that remained stable at the subsequent X-rays (Figure 1b,c).

Although the lesion was not present in chest X-rays made during the first hospitalization, a diagnostic work up was adopted to exclude airway or digestive malformations. The contrast oesophagogram denied the presence of oesophageal duplication/perforation. Considering the possibility of a mediastinal location for a bronchogenic cyst—although it usually appears as a round water-density mass usually located subcarinal, paratracheal, paraesophageal, or para-hilar on chest radiographs [23,24]—a chest CT scan with contrast enhancement was taken. The CT showed a softly marginated and oval-shaped lesion with aerial content (maximum axial and longitudinal sizes of 36 × 23 mm and 40 mm, respectively) in the posterior mediastinum. The lesion, not suggestive of a bronchogenic cyst, minimally compressed the adjacent lung parenchyma and anteriorly dislocated the oesophagus that did not appear in continuity with it (Figure 2a,b). Because the lesion was first detected on chest X-ray taken after the exposure to continuous positive airways pressure (CPAP), and based on the localization in the posterior mediastinum and the absence of tomographic features of bronchogenic cyst and communication with the digestive tract, the diagnosis of retrocardiac pneumomediastinum was formulated. Considering the resolution of asthmatic bronchitis and the wellness of the baby, we adopted the strategy of “wait and see”, and the imaging evaluation at 6 months showed the stability of the size of the air collection on chest X-ray (Figure 3a,b). Obviously, the best end point at 6 months of follow up would be the resolution of the air collection. However, considering the size and the localization of the air collection, which did not make for easy spontaneous drainage, in addition to respiratory wellness, we considered the stability of the lesion to be satisfactory. Based on the patient’s clinical course, serial chest X-rays will be used to monitor the evolution until pneumomediastinum resolution.

## 3. Discussion

We described an uncommon case of retrocardiac pneumomediastinum which occurred in a healthy child after the application of continuous positive pressure through a helmet. Considering the unusual case, we carried out a review of the literature to better understand the causes and management of RP.

While cases of generic pneumomediastinum in young ages are widely described in the literature, we cannot say the same for the retrocardiac one, for which few and old manuscripts are reported. Indeed, the search on Pubmed using a keyword such as “retrocardiac OR posterior pneumomediastinum”; the limits of age, “birth—23 months”; and English language led to 5 results, while the search carried out with a keyword such as “pneumomediastinum” and the same limits led to 762 results, among which, 10 papers described retrocardiac pneumomediastinum [11,12,13,14,15,16,17,18,19,20,21].

In all cases that are summarized in Table 1, the RP occurred in the neonatal period [11,12,13,14,15,16,17,18,19,20,21].

Invasive mechanical ventilation was considered as the primary cause of RP in eight papers [11,12,13,16,18,19,20,21], although Rosenfeld et al. reported other conditions such as hyaline membrane disease (HDM), hypoplastic lung (HL), and meconium aspiration (MA) as causes of RP in neonates [16].

By reviewing the available literature, RP is frequently associated with ventilation-related barotrauma and tracheal and oesophageal perforations [14,15,18,19,20]. In fact, the invasive manoeuvres on the airways/digestive tract which neonates frequently underwent, along with the presence of mediastinal soft tissue, act together to make this group of patients more susceptible to this air leak syndrome. However, unlike what was reported in most of the literature, our case of RP occurred beyond the neonatal period in a child not exposed to invasive manoeuvres.

Although very rare, air leak syndromes such as pneumothorax and pneumomediastinum can complicate the application of non-invasive ventilatory support in patients aged 1 to 18 [25,26,27,28,29]. In fact, Hegde and colleagues described pneumothorax and pneumomediastinum in one child and one adolescent treated during acute respiratory failure with humidified high-flow nasal cannulas (HFNCs) [25]. Similarly, Baudin and colleagues reported their experience on complications associated with the use of HFNCs in a paediatric intensive care unit (ICU). The percentage of new pneumothoraxes in 177 episodes of HFNCs was reported as being around 1%, and no pneumomediastinum occurred in their sample [26].

Hishikawa and colleagues stated the increased use of CPAP via face mask suggested by the update of the Japan Resuscitation Council (JRC) guidelines in 2010 on neonatal resuscitation was linked to a higher prevalence of pulmonary air leak in early-term neonates [27].

Even Hung et al. reported the development of pneumomediastinum after the application of bilevel positive airway pressure (BiPAP) ventilation via a face mask in a 13-year-old male with acute lymphoblastic leukaemia and treated with ventilatory support for post-transplant pneumonitis [28].

More recently, Hazkani and colleagues described a large cohort of 780 paediatric patients affected by obstructive sleep apnoea (OSA) and treated with positive pressure support (PPS) as a bridge after adenotonsillectomy to avoid invasive mechanical ventilation (IMV). In this cohort, only one patient (1%) presented with pneumothorax/pneumomediastinum following PPS usage. The possible mechanism underlying the development of this complication is that high-pressure air flow may potentially disrupt the fascial layer of the neck and result in life-threatening pneumomediastinum and pneumothorax. Nevertheless, the authors concluded that, compared to the non-PPS group, PPS did not appear to cause an increase in rates of pneumomediastinum and pneumothorax. Therefore, PPS use can be considered generally safe and is not associated with increased odds of life-threatening complications [29].

Most of the cases of RP reported in Table 1 described symptomatic leaks with frequent respiratory distress, which habitually resolved spontaneously [11,12,13,14,16,18,19,20,21]. Nevertheless, progressive air collection in the posterior mediastinum has also caused life-threatening situations such as cardiac tamponade [19,21] that required a prompt intervention with surgical drainage. In our case, it is likely that wheezing was caused by asthmatic bronchitis rather than RP. The finding of respiratory well-being and normal chest auscultation despite the radiological persistence of the RP during the follow up period supported this hypothesis.

Because the diagnosis of RP can be challenging since it can be sometimes confused with other conditions (hiatal/diaphragmatic hernia, oesophageal perforation or duplication, or bronchogenic cysts), a chest CT scan can be considered a valuable tool to confirm the suspicion, define the extension, and clarify the presence of contributing factors. As the patient did not undergo tracheal or oesophageal intubation, we did not consider it appropriate to examine the integrity of the trachea or oesophagus through airway or digestive endoscopy.

As summarized in Table 1, conservative treatment and surgical drainage are the two strategy options for the management of air collection in the posterior mediastinum; obviously, the choice of the type of intervention depends on the patient’s clinical condition [18,19,20,21]. In our case, we could adopt a “wait and see” strategy based on stable clinical conditions of our child.

## 4. Conclusions

Although pneumomediastinum is commonly described in young patients as a condition that may arise spontaneously or secondarily, the subtype of retrocardiac pneumomediastinum is less known and often considered to be limited to the neonatal period. We have shared our experience with the hope of increasing awareness of the possible presentation of retrocardiac pneumomediastinum beyond the neonatal age in children not exposed to invasive manoeuvres. In fact, increasing awareness about this uncommon air leak syndrome allows a timely and correct diagnosis and prevents the use of unnecessary diagnostic investigations or invasive therapeutic measures.

## Figures and Tables

**Figure 1 children-10-00649-f001:**
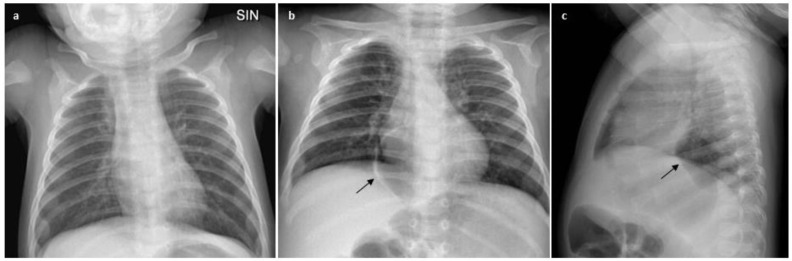
Chest X-ray changes: before (**a**) and after (**b**,**c**) ventilatory support with HelmetCPAP; the oval-shaped lesion is indicated by the black arrow.

**Figure 2 children-10-00649-f002:**
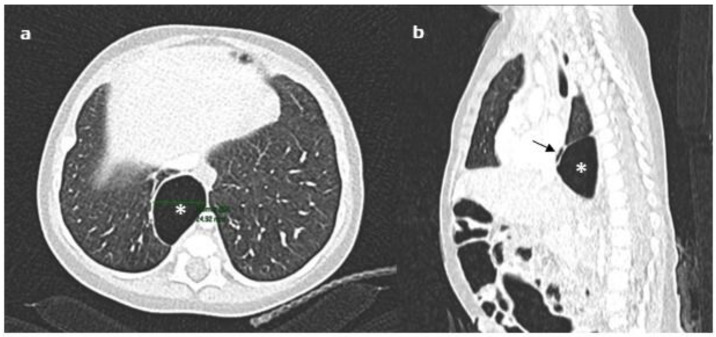
Chest CT scan with contrast enhancement showing the oval-shaped lesion in the posterior mediastinum (indicated by the asterisk): (**a**) the lesion minimally compressed the adjacent lung parenchyma, and (**b**) anteriorly dislocated the oesophagus (black short arrow).

**Figure 3 children-10-00649-f003:**
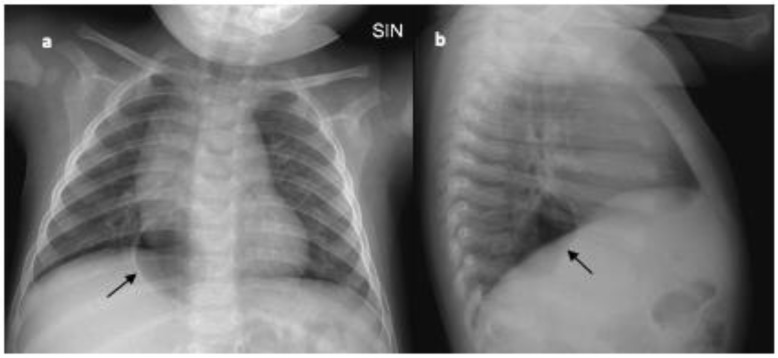
(**a**) Anteroposterior (AP) and (**b**) lateral (L) view of chest X-ray made at 6 months of follow up; the black arrow indicates the unchanged air collection in the posterior mediastinum.

**Table 1 children-10-00649-t001:** Summary table of cases of retrocardiac pneumomediastinum in paediatric population.

Authors	N. Preterm/Term Patients	Total Patients’ n.	Primary Cause	Symptoms	Treatment	Outcomes
Volberg, 1979 [11]	12 pt 2 t	14	14 IMV	RD	n.a.	n.a.
Bowen, 1980 [12]	6 pt	6	6 IMV	RD	n.a.	2 Dead 3 Alive 1 n.a.
Morrison, 1985 [13]	1 t	1	1 IMV	RD	Conservative	Alive
Purohit, 1985 [14]	n.a.	1	n.a.	RD	Conservative	Alive
Amodio, 1986 [15]	2 pt	2	n.a.	n.a.	n.a.	n.a.
Rosenfeld, 1990 [16]	13 pt 1 t	14	1 IMV 12 HMD 1 HL 1 MA	RD	n.a.	6 Dead 8 Alive
Pollack, 1992 [17]	n.a.	n.a.	n.a.	n.a.	n.a.	n.a.
Newman, 1994 [18]	4 pt	4	IMV	RD	2 Drainage 2 Conservative	2 Dead 2 Alive
Kyle, 2011 [19]	1 pt	1	IMV	RD	Drainage	Alive
Beckstrom, 2012 [20]	1 pt	1	IMV	RD	Drainage	Alive
Ponkowski, 2020 [21]	1 pt	1	IMV	RD	Drainage	Alive

Legend: GE, gestational age; pt, preterm; t, term; IMV, invasive mechanical ventilation; HMD, hyaline membrane disease; HL, hypoplastic lung; MA meconium aspiration; RD, respiratory distress; n.a., not available.

## Data Availability

Not applicable.

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
