# Peer review of "Retrocardiac Pneumomediastinum: Description of an Unusual Case and Review of Literature"

_children, 2023, doi:10.3390/children10040649_

Round 1
Reviewer 1 Report
This case report is valuable, making pediatrician aware about this uncommon PR. PR is usually detected at the neonatal stage because of invasive ventilation or invasive manoeuvres. In this case, the PR is not at the neonatal stage nor because of invasive ventilation or invasive manoeuvres. The cause of this case is “helmet continuous positive airway pressure (HCPAP)”. Clinicians may make a wrong diagnosis if they are not aware of the condition.
Minor
1. In the abstract, may the authors delete “introduction”, “case description”, and “conclusion” to make the manuscript more readable.
2. The article type may be changed to “case report”, not “review”
3. Please move the fig legend to the bottom of the fig.
4. Please define 12pt, et al, it’s unclear what’s the meaning of 12 preterm?
5. Please define “CT” when it first appears
6. Please make consistency, line 9 “retrocardiac pneumomediastinum”, and line 20 “inferior pneumomediastinum”, if the authors mean that they are the same.
Author Response
Dear Reviewer, thank you for your appreciation and your comments that allow us to improve the manuscript.
Minor
- In the abstract, may the authors delete “introduction”, “case description”, and “conclusion” to make the manuscript more readable.
We modified the abstract according to your suggestions.
- The article type may be changed to “case report”, not “review”.
We edited the type of manuscript, as correctly indicated by you.
- Please move the fig legend to have modified, as correctly indicated by you the bottom of the fig.
Corrections have been made.
- Please define 12pt, et al, it’s unclear what’s the meaning of 12 preterm?
We added the definition of term (> 37 weeks of gestational age) and preterm (<37 weeks of gestational age) in the text and we modified the second and third column of Table 1, hoping now it is clearer.
- Please define “CT” when it first appears.
We made the suggested correction in the text.
- Please make consistency, line 9 “retrocardiac pneumomediastinum”, and line 20 “inferior pneumomediastinum”, if the authors mean that they are the same.
We made the suggested correction: indeed, “inferior” is not synonymous of “posterior”, as the inferior mediastinum is divided in anterior, middle and posterior mediastinum.
Reviewer 2 Report
This is a very interesting case report on a rare complication of non-invasive ventilation in an infant. Pneumomediastinum is an extremaly rare contition and the retrocardiac location of the air collection is even more so, therefore the authors must be praised for their effort to present such a case to the wider audience. The papaer is well written and includes the discussion on other case serirs of pneumomediastinum in children.
I reccomend it for publication after English language editing
Author Response
Dear Reviewer, thank you for your positive comment. We revised English language as suggested.
Reviewer 3 Report
This case report describes an uncommon case of retrocardiac pneumomediastinum occurring in a 5-months male infant after the application of continuous positive pressure through the helmet. Overall, the paper is interesting and well-written. It underlines the need for awareness of the possible presentation of retrocardiac pneumomediastinum in this population.
I have two comments: First of all, I think that the introduction is rather long. Furthermore, the authors adopted the strategy of “wait and see” and claim that the imaging evaluation at 6 months showed the stability of the air collection. Why don’t they present the relevant chest x-ray? And finally, from a therapeutic point of view, is stability of the lesion a satisfactory end point of the 6 month follow up or should we have anticipated a significant decrease of the air collection’s dimensions? I think that the authors should comment on that.
Author Response
Dear Reviewer, thank you for your comments.
We are aware that the introduction can result too long. Really, we included complete information on the addressed topic (causes, pathogenetic mechanism, symptoms, diagnostic tests, treatment and prognosis) just because our aim was to present this uncommon clinical entity to the wider pediatric audience (who does not necessarily have expertise in the matter). Anyway, if you think that this choice can affect the quality of the manuscript, we are available to update this section.
As you suggested, we added the chest x-ray made after 6 months of follow up.
"Obviously, the best end point at 6 months of follow up would be the resolution of the air collection. Anyway, considering the size and the localization of the air collection which does not make easy the spontaneous drainage, in addition to the patient’s respiratory wellness, we considered the stability of the lesion to be satisfactory. Certainly, the child will continue the clinical and radiological follow-up to define the evolution of RP".
Round 2
Reviewer 3 Report
The authors have addressed my remarks in a thorough and satisfactory fashion.